# Tension can directly suppress Aurora B kinase-triggered release of kinetochore-microtubule attachments

Anna K. de Regt[1], Cordell J. Clark [1], Charles L. Asbury[2✉] & Sue Biggins [1✉]

Chromosome segregation requires sister kinetochores to attach microtubules emanating from opposite spindle poles. Proper attachments come under tension and are stabilized, but defective attachments lacking tension are released, giving another chance for correct attachments to form. This error correction process depends on Aurora B kinase, which phosphorylates kinetochores to destabilize their microtubule attachments. However, the mechanism by which Aurora B distinguishes tense versus relaxed kinetochores remains unclear because it is difficult to detect kinase-triggered detachment and to manipulate kinetochore tension in vivo. To address these challenges, we apply an optical trapping-based assay using soluble Aurora B and reconstituted kinetochore-microtubule attachments. Strikingly, the tension on these attachments suppresses their Aurora B-triggered release, suggesting that tension-dependent changes in the conformation of kinetochores can regulate Aurora B activity or its outcome. Our work uncovers the basis for a key mechano-regulatory event that ensures accurate segregation and may inform studies of other mechanically regulated enzymes.

---

[1] Howard Hughes Medical Institute, Division of Basic Sciences, Fred Hutchinson Cancer Research Center, Seattle, WA, USA. [2] Department of Physiology and Biophysics, University of Washington, Seattle, WA, USA. ✉email: casbury@uw.edu; sbiggins@fredhutch.org

It has been known for decades that tension signals the proper attachment of chromosomes to spindle microtubules and causes selective stabilization of these attachments[1,2], but how tension suppresses the destabilizing activity of Aurora B remains unclear[3–10]. A popular explanation has been that tension-dependent stretching of chromosomes might spatially separate the prominent inner-centromeric pool of Aurora B from kinetochores, thereby inhibiting their phosphorylation[4,11]. However, the inner-centromeric localization of Aurora B is dispensable for accurate chromosome segregation and additional kinetochore binding sites for the kinase have recently been uncovered[12–17], suggesting that other mechanisms might underlie error correction. An alternate 'substrate conformation' model proposes that the structural changes kinetochores undergo when they come under tension block Aurora B phosphorylation[13–15,18–21]. Electron microscopy, fluorescence microscopy, and biochemical cross-linking studies have demonstrated that when kinetochores come under tension they undergo large structural changes[19,22,23], which could potentially block Aurora B phosphorylation via substrate occlusion or another mechanism. This idea is similar to known instances where tension-dependent conformational changes regulate kinase enzymes in other cellular contexts[24–27].

It has not been possible to rigorously test models for Aurora B regulation within cells for several reasons. Aurora B has additional mitotic roles, so pharmacological or genetic inhibition of the kinase acts globally and not specifically on error correction[28,29]. Moreover, it is challenging to test various models in vivo because unattached kinetochores also lack tension, and kinetochores that make attachments quickly come under tension. Finally, the phosphorylation state of individual kinetochores in vivo cannot be precisely correlated with kinase activity because substrate phosphorylation represents a balance of both kinase and opposing phosphatase activity[30–33]. Here, we overcome these challenges using a reconstituted system that allows direct observation of Aurora B-triggered detachments at different levels of tension, applied with a feedback-controlled optical trap. Strikingly, we find that tension on kinetochores inhibits soluble Aurora B from detaching kinetochores from microtubules. Together, our work demonstrates that tension is sufficient to directly suppress the outcome of Aurora B activity.

## Results

**Optical trap assay for Aurora-triggered kinetochore release.** A key prediction of the substrate conformation model is that kinetochore-microtubule attachments should be maintained at high tension even in the presence of excess soluble Aurora B. In contrast, if tension must spatially separate the kinase from its substrates in order to suppress its detachment activity, then the addition of free kinase (i.e., not tethered to the inner centromere) should always promote kinetochore detachment, irrespective of the level of tension. To distinguish these possibilities, we sought to establish an assay in which individual, reconstituted kinetochore-microtubule attachments can be placed under high or low tension using a laser trap, subsequently exposed to soluble active Aurora B kinase, and then monitored until detachment (Fig. 1a). Our previous reconstitution using native kinetochores isolated from budding yeast served as a foundation[34]. We also required a source of purified, soluble Aurora B kinase with sufficient activity to quickly phosphorylate its substrates before a kinetochore would typically detach from a microtubule. Aurora B is a member of the Chromosomal Passenger Complex (CPC) and is activated by the C-terminus of the CPC component INCENP[35].

We made a highly active recombinant Aurora B kinase lacking the inner centromere targeting domains by fusing the budding yeast INCENP[Sli15] activation box (residues 624-698) to the N-terminus of yeast Aurora B[Ipl1], creating a construct we call AurB*. We confirmed its activity by incubating purified kinetochores with ATP-γ-$^{32}$P and either AurB* or a kinase-dead mutant (AurB*-KD), in which the catalytic lysine was replaced with arginine (Fig. 1b). Kinetochores purified from wild-type budding yeast cells lack endogenous Aurora B kinase, and also the inner centromere, but they co-purify with endogenous Mps1 kinase[34,36]. Therefore, we used kinetochores from a temperature-sensitive mutant *mps1-1* strain that had been shifted to the non-permissive temperature to inactivate Mps1. Consistent with previous findings[37], the Mps1-1 kinetochores lacked co-purifying Dam1 complex (Dam1c; Supplementary Fig. 1a), but otherwise resemble wild-type purified kinetochores[36]. In addition, the kinetochores were purified in the presence of irreversible phosphatase inhibitors so we could directly monitor the effects of Aurora B independently of any opposing phosphatase activity. We found that AurB* autoactivated (via autophosphorylation) and efficiently phosphorylated known kinetochore substrates Ndc80, Spc105 and Dsn1[36] (Fig. 1b). The key microtubule-coupling substrate Ndc80 was phosphorylated to greater than 50% completion in under one minute with 0.2 μM AurB* (Fig. 1b and Supplementary Fig. 1b). In contrast, kinetochores incubated with AurB*-KD or only ATP-γ-$^{32}$P exhibited little phosphorylation, confirming the lack of endogenous kinase activity. These experiments show that AurB* can rapidly phosphorylate kinetochores.

**Phosphorylation at physiological sites weakens affinity.** To test whether AurB* phosphorylates physiological targets, we repeated the ATP-γ-$^{32}$P experiment using phospho-deficient mutant kinetochores with alanine substitutions at the seven known Aurora B target sites on the Ndc80 N-terminal 'tail' (Ndc80-7A)[38]. Because combining the *ndc80-7A* and *mps1-1* mutations in budding yeast is lethal, we used kinetochores isolated from strains containing wild-type Mps1 for this experiment. Mps1 also phosphorylates Ndc80[39], so Ndc80 was partially phosphorylated even in the absence of exogenous AurB*, along with other kinetochore proteins (Fig. 2a). However, the addition of AurB* increased phosphorylation on wild-type kinetochores compared to mutant Ndc80-7A kinetochores (Fig. 2a). Thus, AurB* phosphorylates kinetochores on physiologically-relevant sites.

To test whether AurB* is sufficient to reduce kinetochore-microtubule affinity, we purified Mps1-1 kinetochores, retained them on the purification beads, and then incubated the beads with taxol-stabilized microtubules together with either AurB* or AurB*-KD. We subsequently added buffer or ATP, washed the beads, and quantified the bound microtubules by immunoblotting (Fig. 2b). As expected, kinetochores bound to microtubules in the absence of ATP (Fig. 2c, first and second lanes). When ATP was added, there was an AurB*-dependent release of the microtubules (Fig. 2c, lanes 3 through 5) that occurred more slowly if phosphorylation of Ndc80 was blocked at the Aurora B phospho-sites (Supplementary Fig. 2)[38]. These observations show that phosphorylation by AurB* is sufficient to release taxol-stabilized microtubules from kinetochores and confirm that one or more of the known phospho-sites contributes to this release.

**Aurora B activity releases kinetochores from dynamic tips.** To determine whether Aurora B can release kinetochores from dynamic microtubule tips, which are the physiological substrate for error correction, we linked Mps1-1 kinetochores to poly-styrene microbeads and then used an optical trap to attach the beads individually to the assembling tips of single, coverslip-anchored microtubules[34,40,41]. We included native purified Dam1c in the assay buffer to ensure the Mps1-1 kinetochores

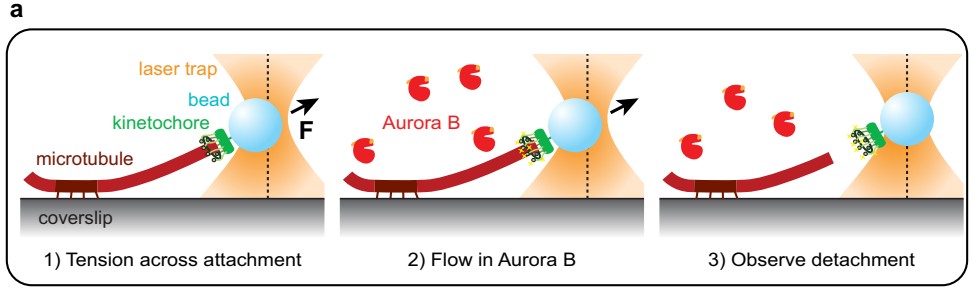

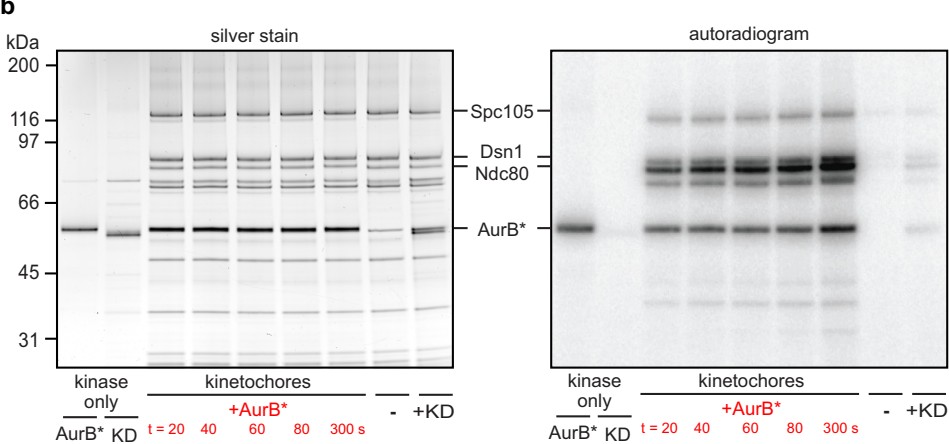

**Fig. 1 Optical trap flow assay for testing whether tension affects Aurora B-triggered release of kinetochore-microtubule attachments. a** Schematic of the assay. A kinetochore-decorated bead is attached to the growing tip of a microtubule anchored to a coverslip. Tension is applied with an optical trap, and free kinase is introduced by gentle buffer exchange. Tip-tracking of the kinetochore-bead is monitored until detachment or interruption of the experiment. **b** An engineered Aurora B construct phosphorylates kinetochores rapidly. Mps1-1 kinetochores (purified from SBY8726) were incubated with either 0.2 µM active AurB* or 0.2 µM kinase-dead mutant AurB*-KD in the presence of $^{32}$P-γ-labeled ATP and then visualized by silver stain and autoradiography. The first two lanes show AurB* or AurB*-KD alone with no kinetochores after 5 min of incubation, the next five lanes show a time course of kinetochores incubated with AurB* for the indicated durations, and each of the last two lanes show kinetochores incubated with either no kinase or AurB*-KD for 3.5 min. The relevant Aurora B substrates are labeled and the positions of protein standards (kDa) are shown on the left. Ndc80 incorporation of $^{32}$P is quantified in Supplementary Fig. 1b. Source data are provided as a Source Data file.

could remain persistently attached to growing and shortening tips under a variety of tensile forces (Supplementary Fig. 3)[34,42]. We adapted the trapping assay to introduce AurB* by flowing kinase-containing buffer into the reaction chamber after a kinetochore-microtubule tip attachment had been established and placed under tension (Figs. 1a and 3a). This approach was technically challenging. Unlike previous versions of our optical trap assay, only one kinetochore attachment could be probed per slide, because any free-floating unattached kinetochores became phosphorylated once AurB* was introduced and thus were unsuitable for further measurements. Moreover, flow in the chamber sometimes caused extra beads or aggregated proteins to catch in the laser, ending the events prematurely (Supplementary Fig. 4a). We therefore made a number of modifications to the assay in order to improve data collection efficiency, including increasing the density of kinetochores on the beads, which avoided lengthy searches for active beads. We applied a constant, relatively low tension of ~1 pN between a kinetochore-coated bead and a microtubule[34], introduced AurB*-KD or AurB* at one of two different concentrations, and then monitored the bead until either detachment or interruption. Tip-tracking of the kinetochore-beads was usually undisturbed by the onset of flow (Fig. 3a and Supplementary Figs. 4a and 4b). The spontaneous detachment rate when the control AurB*-KD was flowed in was $2.3 \pm 0.9\,\mathrm{h^{-1}}$ and this was elevated with active AurB* in a concentration-dependent manner, either two-fold to $4.6 \pm 1.4\,\mathrm{h^{-1}}$ (0.5 µM AurB*) or three-fold to $7.2 \pm 2.2\,\mathrm{h^{-1}}$ (5 µM AurB*) (Fig. 3b and Supplementary Fig. 4c,

d). Analyzing the detachment rates specifically from shortening versus growing microtubule tips showed that the introduction of active AurB* led to more frequent detachment regardless of the tip state (Fig. 3c, d). Together, these data show that AurB* activity is sufficient to detach kinetochores from dynamic microtubule tips that are either growing or shortening in vitro.

**Tension suppresses Aurora B-triggered detachment.** To test the effect of tension on Aurora B-triggered detachment, we sought to repeat the AurB* flow experiment at two different levels of laser trap force, 1.5 and 5 pN, chosen to span the estimated range of physiological forces on yeast kinetochores in vivo[9,43]. During our initial low-force experiments that used beads densely decorated with Mps1-1 kinetochores, spontaneous detachments from shortening microtubules were unusually rare compared to our previous work (Supplementary Fig. 5a), which used beads decorated much more sparsely with wild-type kinetochores[34,40,41]. Lowering the density of Mps1-1 kinetochores on the beads restored the higher detachment rates (Supplementary Fig. 5b), suggesting that at high densities, multiple kinetochores could share the applied load (Supplementary Fig. 5c). Because such load-sharing will reduce the amount of tension per kinetochore, potentially to a level insufficient to inhibit Aurora B, we reverted to a sparse decoration of kinetochores on the beads for all subsequent experiments. Active beads were more difficult to find, as expected, but the Mps1-1 kinetochores in the presence of Dam1c showed a catch bond-like increase in attachment stability with force, consistent with our previous studies using

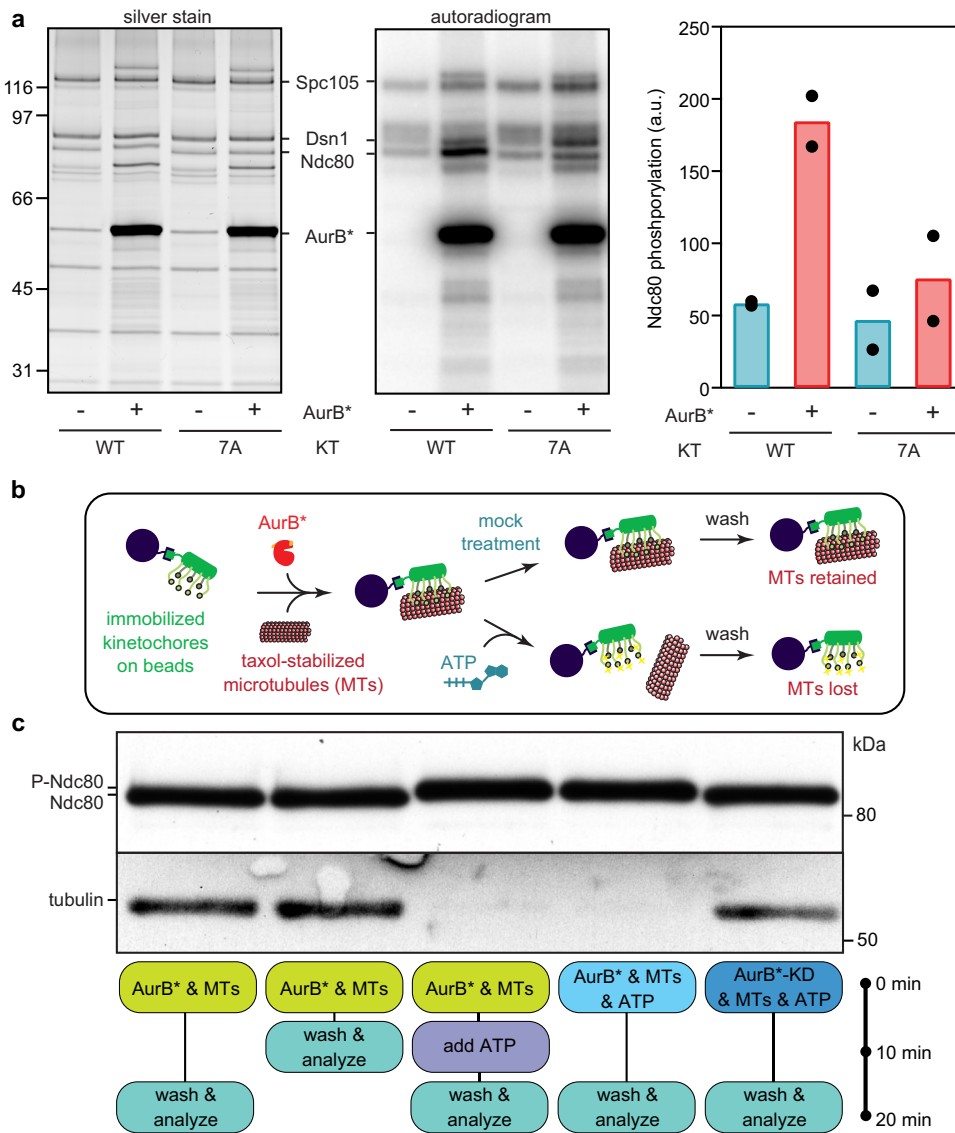

**Fig. 2 Aurora B phosphorylates physiological substrates and reduces kinetochore-microtubule affinity. a** Wild type (WT, SBY8253) or phospho-deficient mutant Ndc80-7A (7 A, SBY8522) kinetochores (KTs) were incubated with either buffer or 1 µM AurB* in the presence of $^{32}$P-γ-labeled ATP for 3 min. A decrease in phosphorylation of the Ndc80 band when comparing lanes 2 and 4 shows that AurB* phosphorylates Ndc80 on one or more of the seven alanine-substituted residues in the phospho-deficient mutant. Ndc80 phosphorylation is quantified at right. Bars represent means from two independent experiments, which are plotted individually as block dots. Controls using buffer alone are represented by cyan bars while experiments with AurB* added are represented by red bars. **b** Schematic for the experiment shown in (**c**). **c** Purified Mps1-1 kinetochores (SBY8726) immobilized on beads were mixed at the indicated times with taxol-stabilized microtubules (MTs), ATP, and AurB* or AurB*-KD, and then washed and analyzed. Components remaining bound to the beads were separated by SDS-PAGE and analyzed by immunoblot using antibodies against Ndc80 (top) or tubulin (bottom). P-Ndc80: phosphorylated Ndc80. Source data are provided as a Source Data file.

wild-type kinetochores[34,41] (Supplementary Fig. 5d). To measure the effect of tension on Aurora B-triggered release, we established kinetochore-microtubule attachments at either low (1.5 pN) or high tension (5 pN), flowed in either AurB*-KD or AurB*, and monitored the detachment rates. AurB* activity strongly promoted kinetochore release at low force, elevating the detachment rate from $10.4 \pm 3.3\,h^{-1}$ in kinase-dead controls up to $28.9 \pm 5.3\,h^{-1}$ when active AurB* was introduced (Fig. 4), a roughly three-fold increase similar to what we observed earlier using densely decorated beads at low force. However, at high force, the detachment rates in the presence of AurB* versus AurB*-KD were statistically indistinguishable, and both around $7\,h^{-1}$. These observations show that tension on the reconstituted kinetochore-microtubule attachments suppressed their release by AurB*.

## Discussion

A number of models have been proposed to explain how Aurora B selectively weakens kinetochore-microtubule attachments that lack tension, while leaving load-bearing, tip-coupled kinetochores unmodified[13–15,18–21]. Rigorously testing such models requires independent control of tension, attachment, and enzyme activity. Here, we developed an assay that fulfills these needs and allows components to be introduced while kinetochore-microtubule attachments are held continuously under tension. While previous work has shown that pre-phosphorylation reduces the microtubule binding affinity of individual kinetochore components[6,40,44], our results provide the first direct demonstration that Aurora B activity is sufficient to detach kinetochores from dynamic microtubule tips and that

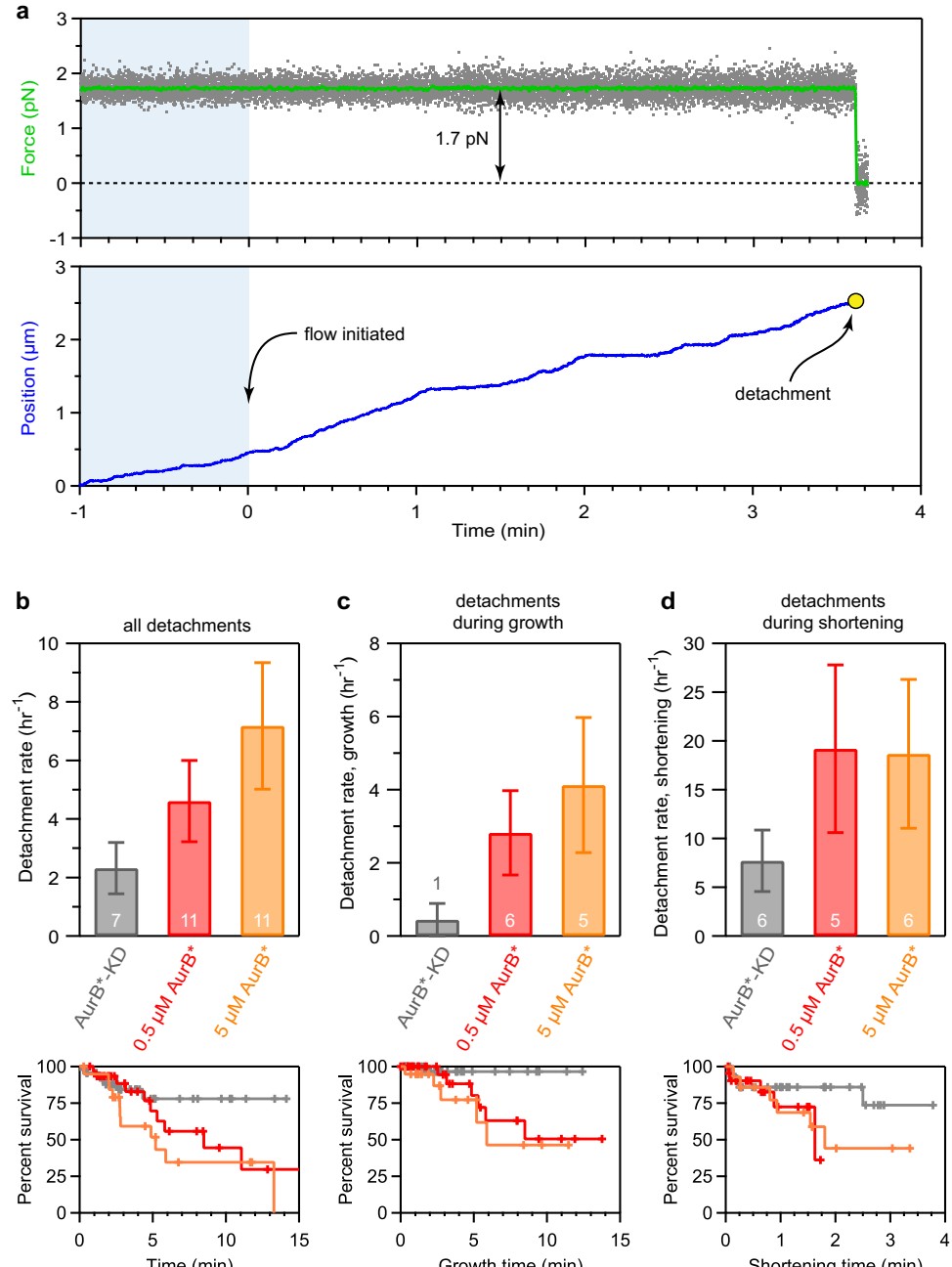

**Fig. 3 Aurora B activity is sufficient to release kinetochores supporting low tension from the tips of dynamic microtubules. a** Example record of optical trap data for an individual detachment event. Gray points show the force applied and green trace shows the same data after smoothing. Blue trace shows the relative position of the kinetochore-decorated bead over time. Kinase introduction occurred at time 0; blue shading indicates data recorded before kinase introduction. See Fig. 1a for a schematic of the experiment. **b** Overall detachment rates (upper bar graphs) and Kaplan-Meier survival plots (lower graphs) for kinetochore-decorated beads supporting ~1 pN of tension in the presence of 5 μM AurB*-KD, 0.5 μM AurB*, or 5 μM AurB*. **c, d** Detachment rates (upper bar graphs) and Kaplan-Meier survival plots (lower graphs) measured specifically during microtubule growth (**c**) or shortening (**d**) for the same conditions as in (**b**). Data for (**b–d**) were collected using a high density of kinetochores on the trapping beads (Dsn1:bead ratio, 3,300), using one biochemical preparation of kinetochores, one preparation of Dam1c, and one preparation of AurB*, to eliminate any possible confounding effects due to prep-to-prep variability. Error bars represent uncertainty due to Poisson statistics. Values inside (or above) bars indicate numbers of detachment events for each condition. Tick marks on Kaplan-Meier plots represent censored data from interrupted events that did not end in detachment. Gray coloring indicates controls using kinase-dead AurB*-KD, red indicates experiments with 0.5 μM active AurB* and orange indicates experiments with 5 μM AurB*. Source data, including numbers of detachments, observation times, rate estimates, and statistical comparisons are provided as a Source Data file.

this effect can be suppressed by tension, consistent with recent work suggesting that Aurora B's inner centromere localization is dispensable for error correction in cells[12,13,15]. The flow assay we developed here can be used in the future to test how tension affects other enzymes involved in error correction.

Because our experiments used soluble AurB*, the tension we applied was not directly borne by the kinase. Therefore, the tension must have altered the kinetochores or the microtubule tips in a way that prevented soluble AurB* from triggering detachments. This substrate conformation effect can explain how

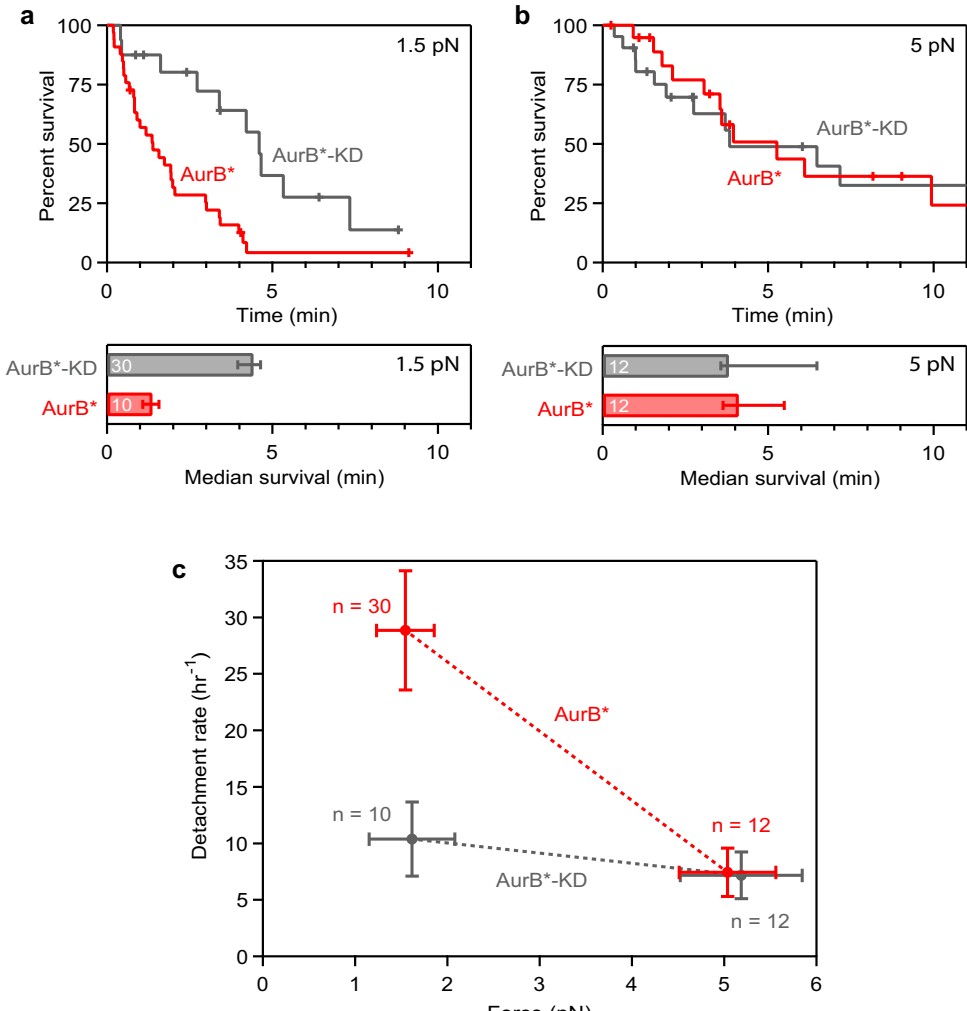

**Fig. 4 Tension can suppress Aurora B-triggered detachment. a**, **b** Kaplan–Meier survival plots (upper graphs) and median survival times (lower bar graphs) for kinetochore-microtubule attachments supporting either ~1.5 pN **a** or ~5 pN of tension (**b**) in the presence of 0.5 μM AurB* or 0.5 μM AurB*-KD. Tick marks on Kaplan–Meier plots represent censored data from interrupted events that did not end in detachment. Based on log-rank tests, p-values for the data shown in (a) and (b) are 0.0016 and 0.98, respectively (kinase-dead versus active AurB*). Median survival bar graphs show times at which the Kaplan-Meier survival probability falls below 50%. Error bars represent interquartile range, estimated by bootstrapping. Values inside bars indicate numbers of detachment events for each condition. (**c**) Overall detachment rates for kinetochore-decorated beads in the presence of 0.5 μM AurB* or 0.5 μM AurB*-KD as a function of the applied force. All data for (**a–c**) were collected using a low density of kinetochores on the trapping beads (Dsn1:bead ratio, 200), using one biochemical preparation of kinetochores, one preparation of Dam1c, and one preparation of AurB*, to eliminate any possible confounding effects due to prep-to-prep variability. Vertical error bars in (**c**) represent uncertainty due to Poisson statistics, based on the numbers of detachment events observed for each condition (n-values, as indicated). Horizontal error bars in (**c**) represent standard deviation. Source data, including numbers of detachments, observation times, rate estimates, and statistical comparisons are provided as a Source Data file. Control data collected using kinase-dead AurB*-KD are colored in gray and data collected using active AurB* are colored in red.

relaxed kinetochore-microtubule attachments are selectively released in vivo while tension-bearing attachments persist[9,10]. Tension causes kinetochores to undergo large structural changes in vivo[19,23,45–47], and such changes could alter the ability of Aurora B to trigger kinetochore-microtubule detachments. We speculate that the key Aurora B substrates might become sterically blocked when a kinetochore comes under tension, or perhaps tension borne by the substrate peptides directly inhibits them from threading into the kinase active site (Fig. 5). It is also formally possible that Aurora B phosphorylation is not affected, per se, and that the tension in our experiments instead suppressed the effects of phosphorylation. However, given that pre-phosphorylated kinetochores failed to bind microtubules (Fig. 2c) and that kinetochores carrying phospho-mimetic substitutions at Aurora B target sites make very weak attachments to

microtubules[39], it seems unlikely to us that Aurora B is able to readily phosphorylate kinetochores at high tension without disrupting their attachments. Regardless of the underlying mechanism of suppression, it must differ from that of the other mechanically regulated kinases uncovered to date, which are all activated rather than inhibited by tension[24–27].

Mechano-regulation of Aurora B via the substrate conformation mechanism does not exclude other tension-dependent effects, such as spatial separation of kinetochores from inner-centromeric Aurora B[4,11], or the intrinsic catch bond-like behavior of kinetochores[34]. While the effect we uncovered here is specific to the kinase because we blocked phosphatase activity, it remains possible that the opposing phosphatase in vivo is also differentially regulated by tension. We suggest that the substrate conformation mechanism works in tandem with these other mechanisms to help ensure

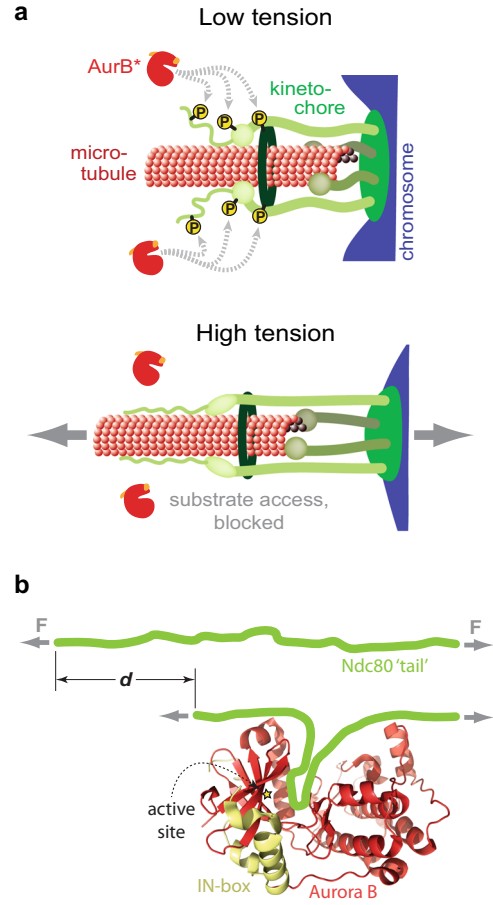

**Fig. 5 Two models illustrating how tension could inhibit Aurora B-triggered detachment. a** The key Aurora B substrate residues might become sterically blocked when a kinetochore undergoes conformational changes caused by tension. **b** In addition, tension borne directly by a substrate peptide might inhibit it from threading into the kinase active site. If a substrate peptide bears a tension $F = 1$ pN, and if threading it into the active site brings its ends closer together by a distance $d = 4$ nm, then its rate of phosphorylation would be reduced ~3-fold (i.e., by $\exp(F \cdot d / k_B T)$, where $k_B T = 4$ pN·nm is thermal energy).

mitotic accuracy. Multiple force-sensitive molecules work together in many contexts, including focal adhesions[24,25,48,49], adherens junctions[50,51], muscle sarcomeres[26,27,52], hemostasis[53,54], and bacterial fimbriae[55–57]. Likewise, a multiplicity of force-sensitive elements probably also underlies the correction of erroneous kinetochore-microtubule attachments during mitosis.

## Methods

**Protein preparation**. The AurB* gene was created in several steps. First, the non-expressed linker between the *SLI15* and *IPL1* genes was removed on two polycistronic vectors expressing the four members of the CPC: pDD2396 (wild-type kinase) and pDD2399 (kinase-dead)[58]. A TEV cleavage site was added between the 6xHis tag and the *IPL1* gene in each plasmid and then the entire *SLI15-IPL1-TEV-6xHis* genes from these constructs were amplified via PCR with BamHI and XhoI sites on the 5′ and 3′ ends, respectively, and inserted into cut pET21b vectors (EMD Biosciences). Each kinase was then expressed as a single polypeptide with a T7 tag on its N-terminus and a 6xHis tag on its C-terminus (plasmids pSB2540 and pSB2894 for wild-type and kinase-dead, respectively). For protein preparation, each plasmid was co-transformed into the Tuner pLys BL21 (DE3) *E. coli* strain (Novagen) with a plasmid that encodes lambda phosphatase[59]. A single transformant was grown to OD 0.9 and expression of both proteins induced with 500 μM IPTG. Protein was expressed for four hours at 30 °C before harvesting. Cells were resuspended in prep buffer (50 mM Tris-HCl pH 7.4, 300 mM KCl, 5 mM MgCl₂, 5 mM ATP, 1 mM DTT, 20 mM imidazole, 10% glycerol) supplemented with 1% Triton X-100, 1 μL Turbo nuclease (Accelagen) and one tablet of Complete EDTA-free protease inhibitor (Roche)[58]

and then lysed via incubation with 0.5 mg/mL lysozyme followed by sonication. Lysate was cleared by centrifugation and then applied to a column of Ni-NTA resin (Qiagen). The column was washed with 10 column volumes (CVs) of prep buffer and then protein-bound resin was resuspended in 3 CVs of prep buffer. DTT was added to 1 mM and 3 mg of TEV protease was added for three hours of room temperature incubation. TEV protease was expressed and purified in house[60]. The resin slurry was re-applied to the empty column and the flow-through was collected along with the flow-through from one wash. This was concentrated and applied to a Superdex 200 FPLC column with storage buffer (50 mM Tris-HCl pH 7.4, 300 mM NaCl, 10% glycerol, 50 mM arginine, 50 mM glutamate). Fractions were analyzed by SDS-PAGE and those from the main peak that eluted at 72 mL were pooled and concentrated.

Kinetochores were purified from budding yeast as previously described with details briefly outlined below[34]. Kinetochore strains have a Dsn1 protein with 6xHis and 3xFlag tags on its C-terminus. Wild-type kinetochores were purified from strain SBY8253; Mps1-1 kinetochores were isolated from SBY8726; Ndc80-7A kinetochores were isolated from SBY8522. Yeast strains were grown asynchronously to OD 4 at room temperature, harvested, washed, and resuspended in Buffer H (25 mM HEPES pH 8.0, 150 mM KCl, 2 mM MgCl₂, 0.1 mM EDTA pH 8.0, 0.1% NP-40, 15% glycerol) supplemented with protease inhibitors, phosphatase inhibitors and 2 mM DTT. SBY8726 for the purification of Mps1-1 kinetochores was shifted to 37 °C for the last two hours of growth. After harvest, yeast drops were frozen in liquid nitrogen and then lysed using a Freezer Mill (SPEX). Lysate was clarified via ultracentrifugation at 98,000 g for 90 min and protein-containing layer was extracted with a syringe. Extract was incubated with magnetic Dynabeads previously conjugated to anti-flag antibodies for two hours at 4 °C and then washed five times in Buffer H. For optical trapping assays, kinetochores were eluted by incubating with 0.83 mg/mL 3xFlag peptide and quantified by comparing the Dsn1 silver-stained band to BSA standards. For enzyme assays, kinetochores were washed but then left attached to beads.

Dam1 complex (Dam1c) was purified via a flag-tagged Dad1 protein from SBY12464 using a similar protocol to the Dsn1-Flag purification described above except the initial immunoprecipitate was first washed three times with a Buffer H with 400 mM KCl before being washed twice in regular Buffer H[61]. The complex was eluted by incubating with 0.83 mg/mL 3xFlag peptide and quantified by comparing the Spc34 silver-stained band to BSA standards.

Taxol-stabilized microtubules were grown by incubation of a mix of unlabeled and Alexa 647-labeled purified bovine tubulin in 1xBRB80 with 4.4% DMSO, 3.2 mM MgCl₂, and 0.8 mM GTP at 37 °C for 30 min. Microtubules and unpolymerized tubulin were separated by ultracentrifugation and sedimented material was resuspended in 1xBRB80 with 10 μM taxol. Stabilized microtubules were used up to three days after polymerization.

**Enzyme assays**. For the radioactive kinase assays, kinetochores were purified and immobilized on magnetic beads and then incubated with AurB*, AurB*-KD, or buffer with 0.2 μM hot (³²P-γ-labeled) and 200 μM cold ATP for the indicated periods of time at room temperature. Experiments were quenched in sample buffer and separated by SDS-PAGE before being silver stained. The gel was dried and exposed to a storage phosphor screen for longer than 48 h for detection of incorporated ³²P.

For the bulk microtubule-binding assays, taxol-stabilized microtubules were polymerized from bovine tubulin for 20 min at 37 °C in the presence of 0.8 mM GTP, 4.4% DMSO, 0.8xBRB80, and 3.2 mM MgCl₂. Polymerized microtubules were sheared using 10 pumps with a syringe and 27 G needle before being sedimented via ultracentrifugation at 98,000 g for 10 min. The supernatant was then removed and the pellet was resuspended in warm 1xBRB80 with 10 μM taxol. For fluorescently-labeled microtubules, eight parts bovine tubulin (10 mg/mL) was mixed with one part Alexa 648-labeled bovine tubulin (~5 mg/mL) for the initial incubation.

For the microtubule binding assays shown in Fig. 2 and Supplementary Fig. 2, the indicated kinetochores were purified and immobilized on beads as described above. For the experiment in Fig. 2c, Mps1-1 kinetochores were then resuspended in 1xBRB80 with 1.8 mg/mL κ-casein, 20 μM taxol, and taxol-stabilized microtubules with or without 500 μM ATP as indicated. After 10 min, either the reaction was washed with 1xBRB80 with 20 μM taxol (BTAX2) and quenched by resuspension in 1x sample buffer (second lane), or 500 μM ATP (central lane) or buffer (other lanes) was added (1 μL into 20 μL reaction). After 20 min from initial resuspension, all experiments were washed with BTAX2 and quenched. Components were separated with SDS-PAGE and assayed with immunoblot. For the microtubule binding assays shown in Supplementary Fig. 2, wild-type or Ndc80-7A kinetochores were incubated for 5 min in 1xBRB80 with 1.8 mg/mL κ-casein, 20 μM taxol, 500 μM ATP, and taxol-stabilized fluorescently-labeled microtubules. At $t = 0$, 1 μM AurB* or buffer was added (1.1 μL into 60 μL reaction). We used fluorescent microtubules so we could precisely quantify microtubules retained on the bead-bound kinetochores by scanning the gel for fluorescence. Timepoints were removed at 10 min and 20 min, washed with BTAX2 and resuspended in 1× sample buffer. Components were separated by SDS-PAGE, fluorescence scanned to visualize tubulin, and transferred to a membrane for immunoblotting to visualize Dsn1. *P* values were calculated by comparing means with a two-tailed unpaired *t*-test with Welch's correction.

**Immunoblotting**. For immunoblots, proteins were transferred from SDS-PAGE gels onto 0.22 μM cellulose paper, blocked at room temperature with 4% milk in PBST, and incubated overnight at 4 °C in primary antibody. Antibody origins and dilutions in PBST were as follows: Ndc80 N-terminus polyclonal antibody generated by the Desai lab: 1:10,000; Flag: Sigma M2 F1804 1:3000; tubulin: EMD Millipore YL1/2 MAB1864 1:1000; Dam1: anti-Dam1 serum was custom generated by Pacific Immunology and used at 1:5000[42]. All antibodies were validated. The Anti-Ndc80 N-terminus antibody was generated in Arshad Desai's lab and has been validated using yeast strains where the endogenous Ndc80 protein was epitope tagged with another epitope to confirm that the protein shifted and that the bands matched. We have used the anti-Flag M2 antibody extensively and have validated it in many publications by comparing yeast strain lysates where proteins are Flag tagged or untagged on immunoblots. The anti-Tubulin antibodies were verified by millipore as described on their website (https://www.emdmillipore.com/US/en/product/Anti-alpha-Tubulin-Antibody-clone-YOL1-34,MM_NF-CBL270-I). The anti-Dam1 antibodies were custom antibodies generated by Pacific Immunology and previously verified in our laboratory[42]. Secondary antibodies were validated by the same methods as the primary antibodies as well as with negative controls lacking primary antibodies to confirm specificity. Blots were then washed again with PBST and incubated with secondary antibody at room temperature. Secondary antibodies were α-mouse (NA931), α-rabbit (NA934), or α-rat (NA935) horseradish peroxidase-conjugated purchased from GE Healthcare and used at 1:1000 dilution in 4% milk in PBST. Blots were then washed again with PBST and ECL substrate from Thermo Scientific used to visualize the proteins. Uncropped and unprocessed scans of gels and blots are provided in the Source Data file.

**Optical trapping**. The optical trapping used polystyrene beads (560 nm average diameter for high density experiments and 440 nm average diameter for low density experiments) conjugated to anti-His antibodies that were incubated with Mps1-1 kinetochores. For the experiments conducted at high density, we incubated the beads at a concentration of 9.0 nM Dsn1 for 1 h at 4 °C, which is >10-fold higher concentration than in our previous force-clamping experiments and potentially resulted in multiple kinetochore particles binding to a single microtubule. For the experiments conducted at low density, we incubated beads (440 nm average diameter) conjugated to anti-His antibodies at a final concentration of 4.0 pM with Mps1-1 kinetochores at a final concentration of 3 nM Dsn1 for 1 h at 4 °C. Reaction chambers were constructed on glass slides using sodium hydroxide- and ethanol-cleaned coverslips for an approximately 14 μL volume chamber. Coverslip-anchored microtubules were grown via successive introduction of: 15 μL of 10 mg/mL biotinylated bovine serum albumin, 50 μL of BRB80 (80mM K-PIPES, 1 mM MgCl$_2$, 1 mM EGTA), 25 μL of 1 mg/mL avidin DN, 50 μL of BRB80, 50 μL of GMPCPP-stabilized microtubule seeds, 100 μL of GTP- and κ-casein-containing BRB80, and then 50 μL of a final reaction buffer consisting of BRB80, 0.3 mg/mL κ-casein, 1 mM GTP, 0.8 mM DTT, 48 mM glucose, 1.6 mg/mL glucose oxidate, 0.3 mg/mL catalase, 3 nM purified Dam1c, 400 μM ATP, bead-bound kinetochores, and 2.6 mg/mL unpolymerized tubulin. This is similar to previous versions of the optical trap assay[34,40,62] except with lower casein and with the addition of ATP, as detailed above. For these experiments, free beads were attached to coverslip bound microtubules using a lower trap stiffness of either 0.028 pN/nm for experiments conducted at 1.5 pN or 0.055 pN/nm for experiments conducted at 5 pN. Beads were dragged to the dynamic microtubule tip under a preload force of 1 pN. Once at the tip, the force was increased to either 1.5 pN or 5 pN. After one minute of tip tracking at either high or low force, 20 μL of flow buffer was introduced via capillary action. Flow buffer contained BRB80, 0.3 mg/mL κ-casein, 1 mM GTP, 0.8 mM DTT, 48 mM glucose, 1.6 mg/mL glucose oxidate, 0.3 mg/mL catalase, 500 μM ATP, 0.5 μM AurB* or AurB*-KD. Flow in the chamber occurred immediately and was observed in the DIC view on the microscope. Events were observed without interference until either a rupture of the attachment occurred, the event was ended by a second bead or large piece of debris falling into the trap, the bead sticking to the coverslip, or by the bead tracking permanently back to the microtubule seed. Detachment rates were calculated by dividing the total number of detachments observed under a given condition by the total attachment time recorded. To avoid biasing the data towards higher detachment rates, we did not discard events that were interrupted and instead calculated overall detachment rates by dividing the number of detachments observed for a given condition by the total observation time after kinase flow was initiated. In all cases that describe error as "due to Poisson statistics," uncertainties were estimated by dividing the detachment rate by the square root of the number of detachments. The rupture force assay for Supplementary Fig. 3a was performed under conditions identical to those previously described[41]. Briefly, the optical trap was used to apply a force of ~2 pN in the direction of microtubule assembly. Once beads were observed to track with microtubule growth for a distance of ~100–300 nm (to ensure end-on attachment), the applied force was increased at a constant rate of 0.25 pN s$^{-1}$ until bead detachment.

**Yeast strains**. *Saccharomyces cerevisiae* strains used in this study were constructed by standard techniques and are derivatives of W303 (*MATa ura3-1 leu2-3,112 his3-11 trp1-1 can1-100 ade2-1 bar1-1*). SBY8253 contains *DSN1-6His-3Flag:URA3*,

SBY8522 contains *DSN1-6His-3Flag:URA3 ndc80::NAT:ndc80-7A:TRP1*, SBY8726 contains *DSN1-6His-3Flag:URA3 mps1-1* and SBY12464 contains *Dad1-3Flag:TRP1*.

**Statistics**. *P*-values comparing the detachment rates of Figs. 3b–d, 4a–c, and Supplementary Fig. 5a–d are provided in the Source Data file and were computed using the E-test as described[63]. *P*-values for comparing the Kaplan–Meier survival plots of Fig. 4a, b are provided in the legend and were computed using the log-rank test. *P*-values comparing the microtubule binding assays shown in Supplementary Fig. 2 are provided in the legend and were calculated using a two-tailed unpaired t-test with Welch's correction.

**Reporting summary**. Further information on research design is available in the Nature Research Reporting Summary linked to this article.

## Data availability
All source data are provided as an Excel document entitled Source Data, with sheets corresponding to each relevant figure. This file includes full scans of all the gels and blots, from Figs. 1b and 2a, c, and Supplementary Figs. 1a and 2a, and all the raw source data used to generate the rate estimates of Figs. 3b, c, d and 4, and Supplementary Fig. 5. Source data are provided with this paper.

## Code availability
Custom software written in Labview (National Instruments) was used for laser trap instrument control and data collection. We currently run it in Labview 2018 and the source code is publicly available at https://github.com/casbury69/laser-trap-control-and-data-acquisition. Custom software written in Igor Pro (Wavemetrics) was used for laser trap data analysis. We currently run it in Igor Pro 8 and the source code is publicly available at https://github.com/casbury69/laser-trap-data-analysis. Assistance with installation and use of the software is available upon request from C.L.A.

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

## Acknowledgements

We thank Arshad Desai for antibodies and Christian Nelson for experimental help. We also thank Trisha Davis, Sharona Gordon, Harmit Malik, Bill Zagotta and members of the S.B. and C.L.A. labs for critical reading of the manuscript. A.K.D. was supported by postdoctoral fellowship PF-15-139-01-CCG from the American Cancer Society. C.L.A. was supported by a Packard Fellowship 2006-30521 and NIH grants R01GM079373 and R35GM134842. S.B. was supported by NIH R01GM064386 and is also an investigator of the Howard Hughes Medical Institute.

## Author contributions

A.K.D. conceptually designed and performed experiments, analyzed data, and wrote the initial draft of the manuscript; C.J.C. performed experiments, analyzed data and edited the manuscript. C.L.A. and S.B. conceptually designed experiments, analyzed data, and wrote the final manuscript with editorial input from A.K.D.

## Competing interests

The authors declare no competing interest.
