## [Peer Review File · Nature Communications]

Tension can directly suppress Aurora B kinase-triggered release of kinetochore-microtubule attachmentsReviewers' Comments:

Reviewer #1 (Remarks to the Author):

In this brief paper by de Regt, Asbury, and Biggins, the authors ask whether tension applied to a kinetochore-microtubule interface is sufficient to prevent Aurora B kinase-triggered detachments. Specifically, one model for tension-dependent error correction in mitosis is that tension-based conformational changes occur on a stretched (high tension) kinetochore that block Aurora B phosphorylation of the kinetochore. In this paper, the authors test this model ("substrate conformation model") by placing tension on stretched, reconstituted kinetochores in the presence of active Aurora B, and then by evaluating whether this tension, in itself, blocks phosphorylation-dependent kinetochore detachment from the microtubule. This is an important question and an interesting approach, as reconstitution allows for precise control of tension as well as kinase activity.

Overall, this is an interesting paper, and the conclusions are well supported by the data. Following are minor comments and suggestions on the manuscript:

1. Perhaps because of the brevity of the paper, it was difficult to follow the transition from the cartoon in Fig. 2a to the results in Figs. 2b. The manuscript would benefit from a more detailed explanation for how the quantity of bound microtubules was monitored using western blots. For example, how is the microtubule length controlled (or is this important), and what does it mean that the tubulin quantity was evaluated by a "fluorescence scan"? Similarly, why does there appear to be a shift in position for tubulin in the Ndc80-7A band? In order to understand Fig. 2b, it would be helpful to include a better description in the main text.
2. Similarly, it is not clear what "relative microtubule binding" is measuring on the y-axis in Fig. 2C. Rather than interpreting the results, it would seem more appropriate to label this axis based on what exactly is being measured, eg, is this microtubule fluorescence intensity normalized $t=0$ minutes? And what exactly is microtubule fluorescence intensity, and how does this exactly relate to "quantity of bound microtubules"? Was there an experiment to validate the output of microtubule fluorescence intensity to demonstrate that it was directly (and linearly) related to the quantity of bound microtubules?
3. Figure 5 is clear and very interesting. The only minor comment I had was regarding the x axis in panel C – it seems that this is the AurB* concentration - the definition of "activity" seems unclear, as I would anticipate that the activity of each AurB* molecule is similar.

Reviewer #2 (Remarks to the Author):

In their recent manuscript, authors de Regt, Asbury, and Biggins address a fundamental question in cell biology, how improper connections between kinetochores and microtubules are detected. It has been previously shown that phosphorylation by the kinase Aurora B can destabilize such connections in vivo and in vitro, but how the phosphorylation activity is directed only to certain attachment states is still very much unclear. In this manuscript, the authors develop an impressive system to monitor the affect of Aurora B activity specifically on kinetochore-microtubule attachments that are under tension. They then use this system to directly test one of the simplest theories for how different microtubule-attachments states are detected, that kinetochores under tension are refractory to Aurora B kinase activity. These experiments are important, interesting and appear to be performed rigorously. However, there are a few limitations to these experiments that I feel need to be directly addressed before publication.

Major points:

1. The authors use a construct that contains only the Kinase and a small peptide from the Sli15 activation domain (the IN box). It is generally believed that Aurora B acts almost exclusively in the context of the other members of the CPC. Is there a reason why the authors didn't use the whole complex purified using one of the published protocols (Cormier et al. 2013 for example)? It seems quite possible that if tension is somehow occluding the kinase from reaching its substrates, that being in a larger complex could enhance that effect.

2. The reasoning for using the levels of tension ranging from 1-8 pN in this study is unclear. It is typically stated in the literature that the CPC corrects attachments that "lack tension" or are "tensionless". It could be that both low- and high-tension attachments have a similarly decreased (but not zero) phosphorylation rate that is significantly lower than the tensionless state. Therefore testing the model would require a comparison of kinetochores without tension versus kinetochores with low tension to see if they differ in response to the kinase. Is there any way to correlate the detachment rates from the bulk bead assay in Figure 2 and the trap assay in Figures 4 and 5? Is it possible to further decrease the tension in the optical trap?

Alternatively, the authors could provide evidence from the literature that syntelically attached kinetochores are still under tension in the range of ~ 1 pN. Certainly there are some counteracting pushing forces such as those contributed by chromokinesins.

3. The difficulty of the optical trap experiments performed here makes me reluctant to suggest too many additional experiments along those lines. However, the extremely high levels of kinase used in those assays makes me question if the system is saturated past the point where tension would make a difference. It has been demonstrated in vivo that too much Ipl1 kinase activity will destabilize or prevent normal attachments, suggesting that an abundance of kinase activity can overwhelm the tension-sensing mechanism (Munoz-Barrera and Monje-Casas 2014). Furthermore, in figure S4a, it appears that a lower concentration of 0.5 micromolar AurB* has a similar effect on the low-tension kinetochores as the 5 micromolar concentration, but very little change when the force is increased to 5 pN. In fact, the percent increase in attachment rate between 1 pN and 5 pN is basically identical for AurB*KD (2.31 \rightarrow 3.97 = 72%) and 0.5 micromolar AurB* (8.24 \rightarrow 13.9 = 69%). This result appears to match what one might expect if the substrate conformation model is true, with an increased frequency of rupture due to the increased force being counteracted by a decrease in the phosphorylation levels. I think that performing these experiments with a greater range of AurB* concentrations could be very enlightening.

4. On a related note, I disagree with the author's statement about the expected outcome from the optical trap experiment for the substrate conformation model. From the bottom of page 8: "The substrate conformation model predicts that detachment rates should be similarly low, with either AurB* or AurB*-KD, when the attachments are held under high tension." It has been shown in mammalian cells that Aurora B greatly increases the turnover rate of metaphase KT-microtubules that are presumably under high tension (Cimini et al. 2006). This is likely true in yeast as well, although perhaps to a lesser extent. Therefore, the presence of AurB* would be expected to greatly increase the turnover of KT-MT connections that are under tension. However, if the substrate conformation model were true, the destabilizing effect on connections that are not under tension would increase to an even greater extent.

Minor points:

5. It would be especially interesting to see if tension would change the degree of kinetochore phosphorylation by Aurora B under these conditions. However, that is likely not possible at this time.

6. This manuscript provides evidence that previous studies concluding that tension is sensed directly by the kinetochore were likely affected by substoichiometric amounts of Dam1 complex in those purified kinetochores. This is an important finding that would be very interesting to the field and I feel that it should therefore not be buried in a supplementary note.

Overall, I think the technology developed in this study is a huge leap forward and will likely lead to a greater mechanistic understanding of Aurora B activity at kinetochore attachments. Unfortunately, the authors have not provided enough information on why they chose the amounts of force and concentrations of kinase used in this study for me to assess the physiological relevance of these in vitro experiments. Furthermore, some of the data appears to be consistent with the model that the authors claim to provide direct evidence against. These issues would need to be addressed before I could recommend the manuscript for publication.

Reviewer #3 (Remarks to the Author):

De Regt et al. establish an optical trap-based in vitro system to test the substrate conformation model whereby Aurora B senses kinetochore tension via the conformational change of its substrates. Under low tension, the substrate is phosphorylated and microtubules are released, while high tension signifying proper attachment results in maintenance of attachment as Aurora B is unable to phosphorylate its substrates. For this model, Aurora B does not need to localise to the inner centromere and can thus be provided free in solution in the reconstitution experiment. The main finding of the paper is that microtubule release is efficiently triggered by a dominant active construct of Aurora B (but not a kinase dead mutant) and this effect is stimulated rather than suppressed by an increase in tension onto purified yeast kinetochores. Thus the experiments provide evidence against the substrate conformation model. Even though this is a negative result, excluding a model is as valuable as providing evidence for a model and thus the paper will be an important contribution to the field.

However, a weakness of the manuscript is that it remains to be established whether the Aurora B target proteins indeed undergo conformational changes in the experimental setup used. To my knowledge, it is not obvious that 5pN (or 8pN) versus 1pN load on the kinetochore-coupled beads would trigger the relevant conformational changes. The dual colour tagging methods used in cells to observed conformational changes in the kinetochore upon attachment / tension might be applicable also to the optical trapping setup used here to validate conformational changes in Ndc80. Alternatively, it might also be possible to design a mutant that cannot undergo the proposed conformational changes to test whether Aurora B can still induce microtubule detachment.

Further, I am concerned about the discrepancy of wildtype kinetochore behaviour between data shown in figure S4b and previously published data shown in S5. It seems that the extremely low detachment rates observed in this study are not just due to the use of Msp1-1 mutant kinetochores supplemented with an excess of Dam1, but also occurred with the wildtype kinetochores used. It might be that this highlights an issue with previous publications rather than the current manuscript, but the authors might want to get to the bottom of this as finding the opposite relationship between force and detachment rates in their assays might call into question the reproducibility of their data. It might for example be possible that the change of behaviour is not due to the purified kinetochores, but the batch of bovine tubulin used to make microtubules as this could contain different contaminants that could trigger detachments in different ways.

We are deeply grateful to the reviewers for their comments on our original manuscript, which was submitted in 2018. As detailed below, their thoughtful and helpful criticisms led us to carry out an extensive set of additional and quite challenging laser trap experiments, and ultimately to thoroughly rework the manuscript. The result, in our opinion, is a much more impactful study showing that kinetochore tension in fact can suppress Aurora B kinase (contrary to what our original data suggested). Because the manuscript has been so extensively revised, we note that some of the reviewer's comments and concerns about the original manuscript no longer apply. Nevertheless, we have responded to all of their original comments below, explaining either how they have been addressed in the new manuscript, or why we believe they are no longer relevant. We sincerely hope the work will now be considered ready for publication.

Reviewer #1

1. Perhaps because of the brevity of the paper, it was difficult to follow the transition from the cartoon in Fig. 2a to the results in Figs. 2b. The manuscript would benefit from a more detailed explanation for how the quantity of bound microtubules was monitored using western blots. For example, how is the microtubule length controlled (or is this important), and what does it mean that the tubulin quantity was evaluated by a "fluorescence scan"? Similarly, why does there appear to be a shift in position for tubulin in the Ndc80-7A band? In order to understand Fig. 2b, it would be helpful to include a better description in the main text.

We apologize for the difficulty deciphering these gel-based measurements of kinetochore-microtubule affinity. In the revised manuscript, these experiments are reported in Figures 2B and 2C, and in Supplemental Figure S2. To clarify the procedures, we describe them now in more detail, both in the main text (pg. 5-6), and also in the methods (pg. 13). Briefly, for the fluorescence experiment of Supplemental Figure S2, we assembled microtubules from a mixture of unlabeled and fluorescent-labeled tubulin, and then broke them into shorter filaments using shear forces created by passing the suspension quickly through a narrow gauge syringe needle multiple times. After performing the binding reaction, we ran an SDS-PAGE gel and quantified bound microtubules by directly scanning the gel at the appropriate wavelength to detect the fluorescent tubulin. Because all the reactions in each experiment (i.e., all the lanes in Figures 2C, and all the lanes in Supplemental Figure S2A) were performed using the same microtubule preparation, made from the same master mix, the microtubule length distributions were standardized across the reactions. The gel shown now in Supplemental Figure S2A "smiled" as it was run, causing a slight band shift in the right-most lanes. We do not believe this shift reflects any meaningful change in electrophoretic mobility.

2. Similarly, it is not clear what "relative microtubule binding" is measuring on the y-axis in Fig. 2C. Rather than interpreting the results, it would seem more appropriate to label this axis based on what exactly is being measured, eg, is this microtubule fluorescence intensity normalized $t=0$ minutes? And what exactly is microtubule fluorescence intensity, and how does this exactly relate to "quantity of bound microtubules"? Was there an experiment to validate the output of microtubule fluorescence intensity to demonstrate that it was directly (and linearly) related to the quantity of bound microtubules?

Again, we are sorry the reviewer found these gel-based experiments confusing. We believe they are better explained in the revised manuscript. The reviewer is referring specifically here to the experiment now shown in Supplemental Figure S2. The "relative microtubule binding" on the y-axis in panel S2B is indeed tubulin fluorescence intensity, measured in three independent experiments like the one shown in panel S2A, and normalized to $t = 0$ minutes. These details are now stated explicitly in the legend of Supplemental Figure S2. The fluorescence intensity of the tubulin bands in these gels (labeled "fluorescence" in panel S2A) is a direct readout of bound microtubules, because the presence of taxol ensured that all the tubulin remained polymerized, and because any unbound

microtubules were washed away before the SDS-PAGE analysis. The step-by-step procedure used for this experiment is described in the materials and methods (on pg. 13).

3. Figure 5 is clear and very interesting. The only minor comment I had was regarding the x axis in panel C – it seems that this is the AurB* concentration - the definition of “activity” seems unclear, as I would anticipate that the activity of each AurB* molecule is similar.

This particular figure is no longer included in the revised manuscript. Instead, new graphs in Figures 3B – 3D show detachment rates as a function of Aurora B concentration. The term “activity” is avoided and the concentration of AurB* is given in micromolar.

Reviewer #2 (Remarks to the Author):

In their recent manuscript, authors de Regt, Asbury, and Biggins address a fundamental question in cell biology, how improper connections between kinetochores and microtubules are detected. It has been previously shown that phosphorylation by the kinase Aurora B can destabilize such connections *in vivo* and *in vitro*, but how the phosphorylation activity is directed only to certain attachment states is still very much unclear. In this manuscript, the authors develop an impressive system to monitor the effect of Aurora B activity specifically on kinetochore-microtubule attachments that are under tension. They then use this system to directly test one of the simplest theories for how different microtubule-attachments states are detected, that kinetochores under tension are refractory to Aurora B kinase activity. These experiments are important, interesting and appear to be performed rigorously. However, there are a few limitations to these experiments that I feel need to be directly addressed before publication.

We thank the reviewer for their interest in our work, and their careful assessment.

Major points:

1. The authors use a construct that contains only the Kinase and a small peptide from the Sli15 activation domain (the IN box). It is generally believed that Aurora B acts almost exclusively in the context of the other members of the CPC. Is there a reason why the authors didn't use the whole complex purified using one of the published protocols (Cormier et al. 2013 for example)? It seems quite possible that if tension is somehow occluding the kinase from reaching its substrates, that being in a larger complex could enhance that effect.

Our rationale for developing the AurB* construct containing just the kinase and the IN-box is two-fold: First and foremost, we specifically sought to test the substrate conformation model, a key tenant of which is that kinetochore tension regulates the free kinase independently of its localization to the inner centromere (or to other specific sites on or near kinetochores). Recent work has indicated multiple kinetochore binding sites for the CPC, in addition to its canonical localization to the inner centromere. Because our small AurB* construct lacks all of these potential interactions, it allowed us to test specifically for a substrate conformation effect. As noted above, we have now discovered that AurB*-triggered detachments can be suppressed by kinetochore tension, so the concern that the full CPC might be needed to see this effect no longer applies.

A second, technical advantage of AurB* is that its purification from *E. coli* yields significantly more active kinase than we have yet been able to achieve when purifying the full CPC. Measuring kinase-triggered detachment in our laser trap flow experiments absolutely requires sufficient kinase activity to cause detachment more frequently than it would occur spontaneously, in the absence of the kinase (as now explained on pg. 4 of the revised manuscript). The smaller fusion protein is straightforward to purify, producing high yields of very active kinase, which were essential for

developing the trap flow experiment. In the future, we hope to study more complete reconstitutions, including the full CPC.

2. The reasoning for using the levels of tension ranging from 1-8 pN in this study is unclear. It is typically stated in the literature that the CPC corrects attachments that "lack tension" or are "tensionless". It could be that both low- and high-tension attachments have a similarly decreased (but not zero) phosphorylation rate that is significantly lower than the tensionless state. Therefore testing the model would require a comparison of kinetochores without tension versus kinetochores with low tension to see if they differ in response to the kinase. Is there any way to correlate the detachment rates from the bulk bead assay in Figure 2 and the trap assay in Figures 4 and 5? Is it possible to further decrease the tension in the optical trap? Alternatively, the authors could provide evidence from the literature that syntelically attached kinetochores are still under tension in the range of ~1 pN. Certainly there are some counteracting pushing forces such as those contributed by chromokinesins.

In retrospect, we agree that our original manuscript did not clearly explain the rationale behind the range of forces we chose. In the revised manuscript, we now state explicitly that the force range we explored, 1 to 5 pN, matches the estimated physiological range of forces *in vivo* (pg. 7). This range was sufficient, under the updated (low kinetochore density) conditions described in the revised manuscript (and detailed below), to reveal a clear suppression of AurB* by the applied tension.

While we do agree that it might be interesting to explore a wider range of forces, it is technically challenging to perform the laser trap experiments at forces below 1 pN, because the kinetochores under such low tensions often fail to track with the growing microtubule tips, instead lagging behind the tips and converting to a side-bound configuration. In the future, we hope to compare kinase-triggered detachment rates for side-bound versus tip-associated kinetochores; but for the present study, we chose to focus exclusively on tip-associated kinetochores.

3. The difficulty of the optical trap experiments performed here makes me reluctant to suggest too many additional experiments along those lines. However, the extremely high levels of kinase used in those assays makes me question if the system is saturated past the point where tension would make a difference. It has been demonstrated *in vivo* that too much Ipl1 kinase activity will destabilize or prevent normal attachments, suggesting that an abundance of kinase activity can overwhelm the tension-sensing mechanism (Munoz-Barrera and Monje-Casas 2014). Furthermore, in figure S4a, it appears that a lower concentration of 0.5 micromolar AurB* has a similar effect on the low-tension kinetochores as the 5 micromolar concentration, but very little change when the force is increased to 5 pN. In fact, the percent increase in attachment rate between 1 pN and 5 pN is basically identical for AurB*KD (2.31 → 3.97 = 72%) and 0.5 micromolar AurB* (8.24 → 13.9 = 69%). This result appears to match what one might expect if the substrate conformation model is true, with an increased frequency of rupture due to the increased force being counteracted by a decrease in the phosphorylation levels. I think that performing these experiments with a greater range of AurB* concentrations could be very enlightening.

We sincerely appreciate the recognition of the difficulty of these experiments. Because we can now demonstrate a tension-dependent suppression of detachment, using 0.5 μM AurB* kinase under low kinetochore density conditions as described in the revised manuscript, we no longer think the concern about high levels of kinase, or the request to use a different range of kinase is relevant. We agree it was a key concern regarding our original dataset and we appreciate the suggestions.

4. On a related note, I disagree with the author's statement about the expected outcome from the optical trap experiment for the substrate conformation model. From the bottom of page 8: "The substrate conformation model predicts that detachment rates should be similarly low, with either AurB* or AurB*-KD, when the attachments are held under high tension." It has been shown in mammalian cells that Aurora B greatly increases the turnover rate of metaphase KT-microtubules that

are presumably under high tension (Cimini et al. 2006). This is likely true in yeast as well, although perhaps to a lesser extent. Therefore, the presence of AurB* would be expected to greatly increase the turnover of KT-MT connections that are under tension. However, if the substrate conformation model were true, the destabilizing effect on connections that are not under tension would increase to an even greater extent.

In retrospect, we agree that this was confusing and possibly overstated. The revised manuscript no longer includes this statement about the expected outcome of the substrate conformation model.

Minor points:

5. It would be especially interesting to see if tension would change the degree of kinetochore phosphorylation by Aurora B under these conditions. However, that is likely not possible at this time.

We agree that it would be exciting to directly assess how tension affects levels of kinetochore phosphorylation. Unfortunately, this is not technically possible at this time. We are hopeful that in the future, simultaneous optical trapping and single molecule fluorescence might allow the binding of phospho-specific antibodies on individual, tension-bearing kinetochores to be observed directly. We have made some progress toward simultaneous trapping and single molecule fluorescence measurements of kinetochores (Deng and Asbury, *Methods Mol Bio* 2017). However, technical difficulties limit the forces we can apply in our current set up to less than 1 pN and require the use of taxol-stabilized (not dynamic) microtubules. So it is not yet feasible to observe changes in the phosphorylation state of a kinetochore coupled to a dynamic microtubule tip and subject to tension in the piconewton range, which is the most physiologically relevant situation, and the situation we have reconstituted in our laser trap flow experiments with AurB*.

6. This manuscript provides evidence that previous studies concluding that tension is sensed directly by the kinetochore were likely affected by substoichiometric amounts of Dam1 complex in those purified kinetochores. This is an important finding that would be very interesting to the field and I feel that it should therefore not be buried in a supplementary note.

We apologize for the confusion caused by the data shown in Supplemental Figure 5 of our original manuscript, which Reviewer #3 also noted. This figure compared detachment rates for wild type kinetochores, measured in our previous studies, to detachment rates for mutant Mps1-1 kinetochores in the presence of free Dam1 complex. The latter exhibited a strikingly lower detachment rate during microtubule disassembly at low force. At that time (i.e., in 2018 when we submitted our original manuscript), the precise reasons for this difference were unclear, but it could have arisen from any one of three changes in experimental conditions, (1) the addition of free Dam1 complex, (2) the use of Mps1-1 kinetochores instead of wild type, or (3) the use of high densities of kinetochores on the trapping beads. Based on the concerns of both Reviewers #2 and #3, we resolved to find out why the discrepancy occurred.

As described in our revised manuscript (pgs. 7 – 8), new experiments showed that the unusually low detachment rate arose solely from the very high density of kinetochores on the trapping beads. The new data (now shown in Supplemental Figures S5A and S5B) implied that multiple kinetochores on a densely decorated bead can share the load of the laser trap, potentially reducing the force per kinetochore to a level insufficient to suppress Aurora kinase. Once we recognized this possibility, we gritted our teeth, and decided that we needed to repeat all the laser trap flow experiments with a low density of kinetochores on the beads. Many hundreds of hours of laser trap time, spanning more than a year, were required. Ultimately, this effort showed that kinetochore tension can suppress AurB*-triggered detachments (Figure 4). It also showed that the intrinsic catch bond-like behavior that we previously discovered using wild type kinetochores (in the absence of Aurora kinase activity) can be recapitulated using Mps1-1 kinetochores, provided they are bound

sparsely to the trapping beads and supplemented with free Dam1 complex (Supplemental Figure S5D).

Overall, I think the technology developed in this study is a huge leap forward and will likely lead to a greater mechanistic understanding of Aurora B activity at kinetochore attachments. Unfortunately, the authors have not provided enough information on why they chose the amounts of force and concentrations of kinase used in this study for me to assess the physiological relevance of these in vitro experiments. Furthermore, some of the data appears to be consistent with the model that the authors claim to provide direct evidence against. These issues would need to be addressed before I could recommend the manuscript for publication.

We thank the reviewer for their positive comments about the potential usefulness of our new approach. We hope that based on our responses above, and with the new results described in our revised manuscript, our work will now be considered suitable for publication.

Reviewer #3 (Remarks to the Author):

De Regt et al. establish an optical trap-based in vitro system to test the substrate conformation model whereby Aurora B senses kinetochore tension via the conformational change of its substrates. Under low tension, the substrate is phosphorylated and microtubules are released, while high tension signifying proper attachment results in maintenance of attachment as Aurora B is unable to phosphorylate its substrates. For this model, Aurora B does not need to localise to the inner centromere and can thus be provided free in solution in the reconstitution experiment. The main finding of the paper is that microtubule release is efficiently triggered by a dominant active construct of Aurora B (but not a kinase dead mutant) and this effect is stimulated rather than suppressed by an increase in tension onto purified yeast kinetochores. Thus the experiments provide evidence against the substrate conformation model. Even though this is a negative result, excluding a model is as valuable as providing evidence for a model and thus the paper will be an important contribution to the field.

We are grateful for the reviewer's careful reading of our original manuscript, and especially for their constructive criticisms, which motivated us to determine why the detachment rates measured in our original flow experiments using Mps1-1 kinetochores were substantially lower than in our previously published measurements using wild type kinetochores. This effort included an extensive set of new laser trap experiments showing that tension in fact can suppress AurB*-triggered detachment, and led to a complete reworking of the paper.

However, a weakness of the manuscript is that it remains to be established whether the Aurora B target proteins indeed undergo conformational changes in the experimental setup used. To my knowledge, it is not obvious that 5pN (or 8pN) versus 1pN load on the kinetochore-coupled beads would trigger the relevant conformational changes. The dual colour tagging methods used in cells to observe conformational changes in the kinetochore upon attachment / tension might be applicable also to the optical trapping setup used here to validate conformational changes in Ndc80.

We agree that it will be potentially very exciting to combine fluorescence-based measurements of intra-kinetochore distance together with force measurements on individual kinetochores. And as noted above, we have made some progress toward simultaneous laser trapping and single molecule fluorescence measurements of kinetochores (Deng and Asbury, *Methods Mol Bio* 2017). Unfortunately, technical difficulties limit the forces we can apply using our current set up to < 1 pN, so we aren't yet able to directly observe changes in kinetochore conformation over the relevant physiological force range, which for yeast kinetochores is estimated to be about 1 to 5 pN (Chacon et al., 2014; Mukherjee et al., 2019). However, new trapping data in our revised manuscript (Figure 4)

show that forces in this range can suppress kinetochore detachment events triggered by AurB*, which is freely soluble and therefore cannot directly bear any of the applied load. This observation indicates that the applied force must have altered either the kinetochores or the microtubule tips in a way that prevented the soluble kinase from triggering detachments, in direct support of the substrate conformation model (as we also explain in the discussion of the revised manuscript, on pg. 9). We hope in the future that a combined trapping and fluorescence approach will allow direct observation of the underlying conformational changes. But our new trapping data already establish the importance of substrate conformation, which we believe is a significant advance in understanding mechanistically how Aurora kinase distinguishes correct from incorrect kinetochore-microtubule attachments.

Alternatively, it might also be possible to design a mutant that cannot undergo the proposed conformational changes to test whether Aurora B can still induce microtubule detachment.

As explained above, our new trapping data indicate that substrate conformational changes must underlie the suppression of AurB*-triggered detachments, but precisely what elements at the kinetochore-microtubule interface are changing, and what structural changes are specifically necessary for the suppression remain unknown. Some candidate changes are suggested by the prior studies comparing intra-kinetochore distances in metaphase cells versus anaphase cells, and in cells treated with various spindle-altering drugs. We agree that creating mutants designed to suppress these conformational changes and testing them using our laser trap approach could be quite interesting. However, at this time, it would require a large effort even to test whether we had such mutant. We view this as a worthwhile goal for the future.

Further, I am concerned about the discrepancy of wildtype kinetochore behaviour between data shown in figure S4b and previously published data shown in S5. It seems that the extremely low detachment rates observed in this study are not just due to the use of Msp1-1 mutant kinetochores supplemented with an excess of Dam1, but also occurred with the wildtype kinetochores used. It might be that this highlights an issue with previous publications rather than the current manuscript, but the authors might want to get to the bottom of this as finding the opposite relationship between force and detachment rates in their assays might call into question the reproducibility of their data.

We are grateful to the reviewer for pushing us to reconcile this issue. This extremely important concern with our initial manuscript was also brought up by Reviewer #2, and it motivated us to further explore the discrepancy between our newer work and our prior publications. Specifically, our newer work using Mps1-1 kinetochores in the presence of Dam1 complex indicated a strikingly lower detachment rate from disassembling tips at low force when compared with our previous measurements using wild type kinetochores. We suspected that this difference might have occurred because we were decorating our laser trapping beads with a relatively high density of kinetochores, in order to make the tricky AurB* trap flow experiments more tractable. (Working at high density avoids lengthy searches for the active beads, capable of binding microtubules.) Indeed, as detailed above in our response to Reviewer #2, point 6 (and shown in Supplemental Figure S5D of the revised manuscript), additional laser trap measurements across a range of decoration densities confirmed that working at high density enables multiple kinetochores to share the applied load, an effect which probably masked the tension-dependent suppression of AurB*. Once we recognized this possibility, we decided to repeat the entire set of AurB* trap flow experiments at low kinetochore density. This was a tremendous amount of additional effort, requiring hundreds of hours of trap time spanning more than a year. It revealed that tension in fact can suppress the ability of AurB* to detach kinetochores from dynamic microtubule tips (as now shown in Figure 4 of the revised manuscript). It also indicated that the intrinsic catch bond-like behavior we previously discovered using wild type kinetochores (in the absence of Aurora kinase activity) can be recapitulated using mutant Mps1-1 kinetochores, provided they are bound sparsely to the laser trapping beads and supplemented with free soluble Dam1 complex (as is now shown in Supplemental Figure S5D).

It might for example be possible that the change of behaviour is not due to the purified kinetochores, but the batch of bovine tubulin used to make microtubules as this could contain different contaminants that could trigger detachments in different ways.

As detailed above, we found that the change in behavior occurred because we had been decorating our trapping beads with a relatively high density of kinetochores. By now we have observed the intrinsic catch bond-like behavior of isolated kinetochores numerous times, including for Akiyoshi 2010 and Miller 2016, and in unpublished work using kinetochores carrying alanine substitutions on Ndc80 (circa 2013) and meiotic kinetochores (circa 2014), and most recently, using Mps1-1 kinetochores supplemented with free Dam1 complex (Supplemental Figure S5D of the revised manuscript). These observations each used a different bovine tubulin preparation, and show that batch-to-batch differences between our bovine tubulin preparations have little effect on the overall stability of kinetochore microtubule attachments in our *in vitro* assays, or on their sensitivity to tension.

REVIEWER COMMENTS

Reviewer #2 (Remarks to the Author):

In their revised manuscript, authors de Regt, Clark, Asbury, and Biggins now perform extensive additional experimentation to address a key discrepancy between their initial results and results that they have published previously. This discrepancy concerns the tension-dependent stabilization of KT-MT attachments in the absence of CPC activity. The authors now find that KT density on the beads is hugely impactful for the tension-dependence of attachments both in the presence and absence of Aurora B. The lower density of KTs on the beads likely better represents the function of a yeast kinetochore. Strikingly, this completely reverses main conclusion of the paper. They now conclude that destabilization of KT-MT attachments by the CPC is indeed tension-dependent and can occur independently of any currently known CPC localization mechanism. This result supports a model wherein the conformation of Aurora B substrates is sensitive to tension and the change in conformation affects Aurora B destabilization. Overall, the resolution of the previous inconsistency in the manuscript is a massive improvement and resolves many of my former criticisms. Of course, revisions that drastically change the main conclusion of the paper open it up to new critiques as well. I now have some additional concerns and unanswered questions that relate to the changes in the manuscript.

1. For the differences in detachment rates under varying force without Aurora B activity that have been measured previously and in this manuscript, a key contributing factor to the tension-dependence is the proportion of microtubules that are polymerizing vs. depolymerizing during the experiment. In the key experiment (Figure 4), there is no distinction between polymerizing and depolymerizing microtubules. Are detachments from polymerizing and depolymerizing microtubules equally represented in both 1.5 pN and 5 pN of force? Do MTs under high force ever convert to depolymerization? It would be nice to compare these results to the results in Figure 3, but unfortunately the Figure 3 experiments were done under conditions of high bead density, which corresponds to the opposite result for tension-dependent response to Aurora B. If it is impossible to get a sufficient number of detachment events to confidently distinguish between the off rates for polymerizing vs depolymerizing MTs, it would at least be nice to see this data for the few events that have already been recorded.

2. The authors note in the discussion that they cannot currently distinguish between the tension-dependent affect on Aurora B phosphorylation being a result of an inability of the kinase to phosphorylate the kinetochores and an inability of phosphorylated kinetochores to detach from microtubules under high tension. I wonder why they cannot test this by creating a phosphorylated state prior to attachment. This could be done either by preincubating the KTs with AurB* or by using the phosphomimic kinetochore mutations that they have characterized before. With the phosphomimics, they have at least shown before that they are capable of making load-bearing attachments. If there is no decrease in detachment rate upon increased tension in these conditions, then one could conclude that the decrease in detachment rate observed in the presence of Aurora B is due to a lack of phosphorylation. These experiments would presumably be easier than the experiments in Figure 4 since there could be multiple measurement attempts made per slide.

3. The complete reversal of the conclusion of the manuscript based off of a relatively minor change in how the experiments are performed (KT concentration on the beads) highlights the fact that the entire study relies on the results from a single experiment that is extremely difficult, and evidently rather fickle. The author's conclusions are based on a particular concentration of kinase, comparing only two specific amounts of tension, under certain buffer conditions, with a given density of KTs, with one concentration of supplemented Dam1 complex. While I appreciate the efforts of the authors to recapitulate realistic conditions as best they can, it seems like minor changes to any of these parameters has the potential to radically alter the results in unexpected ways. It is therefore difficult to have confidence in the robustness of the main conclusion. However, I do think that the results convincingly show that it is possible for tension to decrease Aurora B induced KT-MT turnover, just as their previous results showed that it is possible for tension to increase Aurora B induced KT-MT turnover. The fact that such outcomes are possible is

very interesting, important to the field, and something that should certainly be further explored in the future. With this in mind, I think that it's important to be precise with the language used to describe the key result. At times, the authors emphasize the limitations of the conclusions that can be drawn from a single experiment. For example, the conclusion at the end of the introduction reads, "tension can directly suppress the outcome of Aurora B activity". I think this statement reflects their results better than the title of Figure 4, which reads, "Tension suppresses Aurora B-triggered detachment." instead of "Tension can suppress Aurora B-triggered detachment." The title could also be changed to be more consistent with the conclusion in the introduction.

4. In figure S5A and S5B, it looks like there are considerably faster detachment rates at a 200 Dsn1:bead ratio than at a 280 Dsn1:bead ratio. This result suggests that even at the lower concentration of KTs used in their assays, there are still multiple KTs contributing to attachments. Given the sensitivity of the assay to KT density on the bead, it seems like this could greatly affect the results of the experiment. I feel that some discussion of this result and its implications on the overall interpretation is warranted.

Reviewer #3 (Remarks to the Author):

The authors very carefully revised the manuscript, which resulted in a change of their main conclusion. I appreciate that the authors carefully assessed the discrepancy between the current study and their earlier work and can now explain how higher kinetochore protein density on the beads caused a very different behaviour than previously published, but also masked the tension effect that they could now confirm. The revised work is succinct but very clearly presented. The methods are detailed and supporting data are provided. Thus I recommend publication of the manuscript and have no further requests for changes.

We are grateful to the reviewers for their thoughtful comments. We address each of their remarks below:

Reviewer #2 (Remarks to the Author):

In their revised manuscript, authors de Regt, Clark, Asbury, and Biggins now perform extensive additional experimentation to address a key discrepancy between their initial results and results that they have published previously. This discrepancy concerns the tension-dependent stabilization of KT-MT attachments in the absence of CPC activity. The authors now find that KT density on the beads is hugely impactful for the tension-dependence of attachments both in the presence and absence of Aurora B. The lower density of KTs on the beads likely better represents the function of a yeast kinetochore. Strikingly, this completely reverses main conclusion of the paper. They now conclude that destabilization of KT-MT attachments by the CPC is indeed tension-dependent and can occur independently of any currently known CPC localization mechanism. This result supports a model wherein the conformation of Aurora B substrates is sensitive to tension and the change in conformation affects Aurora B destabilization. Overall, the resolution of the previous inconsistency in the manuscript is a massive improvement and resolves many of my former criticisms. Of course, revisions that drastically change the main conclusion of the paper open it up to new critiques as well. I now have some additional concerns and unanswered questions that relate to the changes in the manuscript.

We appreciate the reviewer's recognition that we have addressed inconsistencies in our original version and resolved many of their former criticisms. We are particularly grateful for their insights about the original submission that prompted us to explore the change in behavior at high kinetochore density on the beads. We understand that the new conclusion brings up additional questions that we address point-by-point below.

1. For the differences in detachment rates under varying force without Aurora B activity that have been measured previously and in this manuscript, a key contributing factor to the tension-dependence is the proportion of microtubules that are polymerizing vs. depolymerizing during the experiment. In the key experiment (Figure 4), there is no distinction between polymerizing and depolymerizing microtubules. Are detachments from polymerizing and depolymerizing microtubules equally represented in both 1.5 pN and 5 pN of force? Do MTs under high force ever convert to depolymerization? It would be nice to compare these results to the results in Figure 3, but unfortunately the Figure 3 experiments were done under conditions of high bead density, which corresponds to the opposite result for tension-dependent response to Aurora B. If it is impossible to get a sufficient number of detachment events to confidently distinguish between the off rates for polymerizing vs depolymerizing MTs, it would at least be nice to see this data for the few events that have already been recorded.

We did not distinguish detachment rates during growth versus shortening in Figure 4 because the number of detachment events recorded specifically during shortening in that experiment was insufficient to make a statistically meaningful comparison. Recording detachments during tip shortening is very challenging, in part because the experimenter must first wait for a catastrophe event, when the microtubule tip converts spontaneously into depolymerization, which occurs infrequently (usually only once every 20 to 60 min, depending on conditions). Moreover, a kinetochore-decorated bead will often detach prior to catastrophe, during tip growth, precluding any observation of catastrophe or tip shortening. Despite these challenges, we have managed in some experiments to record substantial numbers of detachments during tip shortening (e.g., in the experiments of Figure 3, and in Akiyoshi et al., *Nature* 2010, and Franck et al., *Nat Cell Biol* 2007). But due to the added difficulties associated with flow-based introduction of Aurora B at low kinetochore density, we were only able to record a small number of these events in the experiments of Figure 4. We have added these data into Supplementary Data Table 1, as requested.

2. The authors note in the discussion that they cannot currently distinguish between the tension-dependent affect on Aurora B phosphorylation being a result of an inability of the kinase to phosphorylate the kinetochores and an inability of phosphorylated kinetochores to detach from microtubules under high tension. I wonder why they cannot test this by creating a phosphorylated state prior to attachment. This could be done either by preincubating the KTs with AurB* or by using the phosphomimic kinetochore mutations that they have characterized before. With the phosphomimics, they have at least shown before that they are capable of making load-bearing attachments. If there is no decrease in detachment rate upon increased tension in these conditions, then one could conclude that the decrease in detachment rate observed in the presence of Aurora B is due to a lack of phosphorylation. These experiments would presumably be easier than the experiments in Figure 4 since there could be multiple measurement attempts made per slide.

We are also interested in the question of whether Aurora B no longer phosphorylates kinetochores at high tension, or whether the kinetochores become insensitive to phosphorylation. And we appreciate the reviewer's suggestions for how we might distinguish between these two possibilities. However, because pre-phosphorylated kinetochores do not bind detectably to microtubules (shown in Figure 2B), the proposed experiment would be difficult to perform with pre-phosphorylated kinetochores. We did previously show that kinetochores carrying phospho-mimetic substitutions

at some Aurora B sites can make load-bearing attachments, albeit weakly (Sarangapani et al., *PNAS* 2013), but these kinetochores do not carry substitutions at all of the Aurora B sites. Nevertheless, we did make preliminary lifetime measurements using partially phospho-mimetic kinetochores: Ndc80-7D kinetochores had a maximum lifetime of ~15 min at a force of ~2 pN but fell to near-zero lifetime at 6 pN and Dam1-4D kinetochores were even weaker. Given that fully phosphorylated and partially phospho-mimetic kinetochores have such a difficult time binding and remaining coupled to microtubules, it seems unlikely to us that Aurora B could phosphorylate kinetochores at high tension without disrupting their attachments. We have added this point to the discussion (p. 10, lines 4-8).

3. The complete reversal of the conclusion of the manuscript based off of a relatively minor change in how the experiments are performed (KT concentration on the beads) highlights the fact that the entire study relies on the results from a single experiment that is extremely difficult, and evidently rather fickle. The author's conclusions are based on a particular concentration of kinase, comparing only two specific amounts of tension, under certain buffer conditions, with a given density of KTs, with one concentration of supplemented Dam1 complex. While I appreciate the efforts of the authors to recapitulate realistic conditions as best they can, it seems like minor changes to any of these parameters has the potential to radically alter the results in unexpected ways. It is therefore difficult to have confidence in the robustness of the main conclusion. However, I do think that the results convincingly show that it is possible for tension to decrease Aurora B induced KT-MT turnover, just as their previous results showed that it is possible for tension to increase Aurora B induced KT-MT turnover. The fact that such outcomes are possible is very interesting, important to the field, and something that should certainly be further explored in the future. With this in mind, I think that it's important to be precise with the language used to describe the key result. At times, the authors emphasize the limitations of the conclusions that can be drawn from a single experiment. For example, the conclusion at the end of the introduction reads, "tension can directly suppress the outcome of Aurora B activity". I think this statement reflects their results better than the title of Figure 4, which reads, "Tension suppresses Aurora B-triggered detachment." instead of "Tension can suppress Aurora B-triggered detachment." The title could also be changed to be more consistent with the conclusion in the introduction.

We thank the reviewer for understanding the challenges associated with the single molecule experiments. While it may appear that minor changes to parameters radically altered the outcome, we think it is unlikely that our new result using beads decorated with a lower density of kinetochores is an artifact of the altered conditions. When we performed the original experiments, we did not appreciate that a high kinetochore density on the beads could result in load sharing. The concerns from the first round of review prompted us to test this possibility, and we now know that it altered our conclusion. However, our new results at low kinetochore density show that high force can suppress detachments, an outcome that is much more difficult to imagine could arise spuriously from minor changes in experimental conditions. We fully agree that we need to avoid overstating our conclusions, and we have therefore modified the title of Figure 4 as well as the overall title of the paper, as suggested.

4. In figure S5A and S5B, it looks like there are considerably faster detachment rates at a 200 Dsn1:bead ratio than at a 280 Dsn1:bead ratio. This result suggests that even at the lower concentration of KTs used in their assays, there are still multiple KTs contributing to attachments. Given the sensitivity of the assay to KT density on the bead, it seems like this could greatly affect the results of the experiment. I feel that some discussion of this result and its implications on the overall interpretation is warranted.

We appreciate the reviewer pointing out this concern. We agree that some load-sharing might still occur even at the sparsest decoration densities, since detachment rates during shortening remain sensitive to the Dsn1:bead ratio (as shown in Fig. S5B). However, this does not affect the overall conclusion of the experiment because high tension suppressed detachments (Fig. S5D), an observation consistent with our prior single molecule experiments and unlikely to be a spurious effect caused by load-sharing. The possibility that some load-sharing might still occur even at the sparsest decoration densities is now mentioned explicitly in the revised manuscript (in the legend of Fig. S5).

Reviewer #3 (Remarks to the Author):

The authors very carefully revised the manuscript, which resulted in a change of their main conclusion. I appreciate that the authors carefully assessed the discrepancy between the current study and their earlier work and can now explain how higher kinetochore protein density on the beads caused a very different behaviour than previously published, but also masked the tension effect that they could now confirm. The revised work is succinct but very clearly presented. The methods are detailed and supporting data are provided. Thus I recommend publication of the manuscript and have no further requests for changes.

We are grateful to the reviewer for their interest in our work and their thoughtful comments after the first submission that helped us revise the manuscript.

REVIEWERS' COMMENTS

Reviewer #2 (Remarks to the Author):

The authors have thoroughly addressed all of my concerns. I appreciate their considered responses to all of my nitpicks.